# Multiple Roles for Cholinergic Signaling from the Perspective of Stem Cell Function

**DOI:** 10.3390/ijms22020666

**Published:** 2021-01-11

**Authors:** Toshio Takahashi

**Affiliations:** Suntory Foundation for Life Sciences, Bioorganic Research Institute, Kyoto 619-0284, Japan; takahashi@sunbor.or.jp

**Keywords:** muscarinic acetylcholine receptor (mAChR), nicotinic acetylcholine receptor (nAChR), neural stem cell (NSC), hair follicle stem cell (HFSC), melanocyte stem cell (MeSC), intestinal stem cell (ISC), homeostasis, niche

## Abstract

Stem cells have extensive proliferative potential and the ability to differentiate into one or more mature cell types. The mechanisms by which stem cells accomplish self-renewal provide fundamental insight into the origin and design of multicellular organisms. These pathways allow the repair of damage and extend organismal life beyond that of component cells, and they probably preceded the evolution of complex metazoans. Understanding the true nature of stem cells can only come from discovering how they are regulated. The concept that stem cells are controlled by particular microenvironments, also known as niches, has been widely accepted. Technical advances now allow characterization of the zones that maintain and control stem cell activity in several organs, including the brain, skin, and gut. Cholinergic neurons release acetylcholine (ACh) that mediates chemical transmission via ACh receptors such as nicotinic and muscarinic receptors. Although the cholinergic system is composed of organized nerve cells, the system is also involved in mammalian non-neuronal cells, including stem cells, embryonic stem cells, epithelial cells, and endothelial cells. Thus, cholinergic signaling plays a pivotal role in controlling their behaviors. Studies regarding this signal are beginning to unify our understanding of stem cell regulation at the cellular and molecular levels, and they are expected to advance efforts to control stem cells therapeutically. The present article reviews recent findings about cholinergic signaling that is essential to control stem cell function in a cholinergic niche.

## 1. Introduction

The appearance of the nervous system is considered to be an evolutionally epochal event that fundamentally changed how control is achieved within a multicellular body. Recent progress in genomics, molecular phylogenetics, developmental biology, and the study of simple nervous systems in living animals such as Cnidaria has provided a wealth of new empirical information about the earliest stages in neuronal evolution. Ancestral Cnidarians diverged over 500 million years ago in animal evolution [1]. Cnidaria such as *Hydra*, which is a descendant of ancestral Cnidarians, are composed of multiple cell types that represent the fundamental architecture of multicellular organisms. *Hydra* also have multipotent interstitial stem cells, which differentiate into nerve cells [2], nematocytes [2], gland cells [3], and germ cells [4]. The nervous system of *Hydra* is simple and is composed of a nerve net that extends throughout the animal. The cnidarian nervous system is mainly peptidergic [5]. It has been suggested that classical molecules such as acetylcholine (ACh) also contribute to the *Hydra* nervous system from the results of pharmacological experiments [6]. The membrane-bound ACh receptor and acetylcholinesterase have been demonstrated and confirmed by genome analysis [7,8]. Although ACh and other ACh receptor agonists function in neuronal and/or neuromuscular communication to regulate muscle contractions in *Hydra*, ACh itself has not been detected.

ACh is the first substance proven to be a neurotransmitter [9]. ACh is the major parasympathetic mediator and is synthesized by the catalytic conversion of acetyl-CoA and choline to CoA and ACh by choline acetyltransferase (ChAT) (Figure 1) [10,11]. In cholinergic neurons, ACh is transported into synaptic vesicles via the vesicular ACh transporter (VAChT) and stored there until released by exocytosis (Figure 1). VAChT was first cloned and characterized in *Caenorhabditis elegans* [12]. After the release of ACh into the synaptic cleft, the neurotransmitter evokes membrane action potentials by binding to ACh receptors (Figure 1). Then, ACh is rapidly and specifically degraded by acetylcholinesterase (AChE) and butyrylcholinesterase, which is a second, non-specific cholinesterase in mammals that also produces choline and acetic acid (Figure 1) [13,14]. By-products of choline are taken up into the presynaptic side of the synapse via the high-affinity choline transporter and reused for ACh synthesis (Figure 1) [15]. The organic cation, choline, is a substrate for carriers of organic cation transporters (OCTs). To date, three different OCTs (OCT1–3) that transport choline from the extracellular space into nerve cells have been identified [16]. ACh is stored at and released from VAChT in neurons [17]. Of interest, VAChT is expressed in a cell-type specific manner in non-neuronal cells [18]. The cells that do not express VAChT have no ability to store ACh but directly release ACh via OCTs [19,20]. Thus, OCTs are two-in-one choline and ACh transporters. These ACh synthetic pathways described above constitute the cholinergic system.

Schofield [21] originally hypothesized the existence of a microenvironment that is required for the maintenance of stem cells using hematopoietic stem cells and called such a region a “niche”. A niche is considered to be a subset of tissue cells and extracellular substrates that can indefinitely maintain stem cells and control their self-renewal and progenitor cell production in vivo (Figure 2). Specialized internal mechanisms and external signals confer the capacity of growth and differentiation to stem cells such as early embryonic cells in niches. Experimental evidence has revealed that ACh is widely distributed in biological systems beyond the nervous system. The widespread distribution suggests that ACh may be involved in regulation of stem cell functions such as proliferation, differentiation, and the establishment of cell–cell interactions [22]. Thus, the study of cholinergic mechanisms focusing on the regulation of proliferation, differentiation, and maintenance of stem cells is of great interest. Our previous pharmacological studies revealed that ACh is synthesized in intestinal epithelial cells and plays a role in cell division and the differentiation of Leucine-rich repeat-containing G-protein-coupled receptor 5-positive (Lgr5^+^) intestinal stem cells (ISCs) in the small intestine by binding to muscarinic ACh receptors (mAChRs) in crypt-villus organoids [23]. Furthermore, mAChRs and nicotinic ACh receptors (nAChRs) are involved in the proliferation of mouse embryonic and induced pluripotent stem cells [24,25,26]. This evidence leads us to propose the presence of a cholinergic niche that affects stem cell behavior. This review focuses on the multiple roles of cholinergic signaling in stem cells that contribute to extensive regeneration and remodeling in adults, including those present in the brain, skin, and gut.

## 2. Neural Stem Cells (NSCs) in the Adult Mammalian Brain

Adult mammalian neural stem cells (NSCs) contribute to brain plasticity via the generation of new neurons throughout life [27]. Adult NSCs also have fundamental properties of self-renewal, relative quiescence, differentiation capacity, and residence within a specific environmental niche similar to other adult somatic stem cells (Figure 3) [28]. New neurons are derived from NSCs that reside in two major neurogenic niches, the subventricular zone (SVZ) in the lateral ventricles and subgranular zone (SGZ) in the dentate gyrus of the hippocampus [28,29]. In the adult SVZ niche, NSCs give rise to neurons and oligodendrocytes [28]. On the other hand, neurons and astrocytes, but not oligodendrocytes, are generated from NSCs in the adult SGZ [30]. In this section, I review cholinergic signaling involved in postnatal/adult neurogenesis and how patterns of neuronal activity differentially and/or synergistically modulate downstream signaling in NSCs.

### 2.1. Cholinergic Activation of NSCs in the SVZ

Neurogenesis in the SVZ of the olfactory bulb (OB) continues throughout adulthood [31,32]. NSCs in the SVZ generate neuroblasts, which migrate tangentially through the rostral migratory stream toward the OB, and the neuroblasts finally differentiate into interneurons [33]. Within the dentate gyrus and OB, the interneurons abundantly express mAChRs and nAChRs [34], suggesting that the cholinergic system plays a role in regulating neurogenesis.

Accordingly, an in vivo nicotinic exposure experiment was carried out in the SVZ in adult rats, but no effect on proliferation was seen [35,36], suggesting that nAChRs may not be involved in adult OB neurogenesis. However, Mechawar and coworkers [37] used knockout mice to answer the question of whether nAChRs are involved in events downstream of NSC proliferation in the SVZ. They undertook a study of OB neurogenesis using β2^−/−^ mice that were subjected or not subjected to chronic nicotine exposure and found that β2-containing nAChRs are specifically involved in the survival of newborn granule cells in the OB local circuits. Unexpectedly, the behavior of β2^−/−^ mice indicated a less robust short-term olfactory memory than their wild-type (WT) littermates. Furthermore, a pharmacological study using donepezil, a potent AChE inhibitor, revealed that cholinergic stimulation promotes the survival of newborn neurons in the adult OB [38]. Two interesting studies suggested that adult NSCs in the SVZ are regionally specified at an early embryonic stage and then remain largely quiescent until reactivation in the postnatal period [39,40]. The key molecule for postnatal reactivation of SVZ NSCs may be ACh via activation of nAChRs.

### 2.2. Cholinergic Activation of NSCs in the SGZ

Adult hippocampal neurogenesis is tightly controlled by NSCs located in the SGZ of the mammalian dentate gyrus that proliferate, differentiate, are maintained, and integrate into the local circuitry throughout life [41,42,43,44,45]. The cholinergic system is involved in the regulation of adult hippocampal neurogenesis. The dentate gyrus receives input from the basal forebrain through GABAergic and cholinergic projection neurons [46,47]. Injection of fibroblasts secreting ACh into the hippocampus reverses cognitive decline by increasing the proliferation of NSCs [48,49,50]. Furthermore, the administration of an AChE inhibitor increases NSC proliferation and promotes the survival of immature neurons through the α7 nAChR subtype [49,51,52,53].

In the SGZ, pharmacological activation of α7-subunit-containing nAChRs increases cellular proliferation [54]. Homomeric α7 nAChRs contribute to cognition, attention, learning, and memory through fast signal transduction [55,56]. α7 nAChRs have been implicated in diseases including epilepsy, autism, schizophrenia, and Alzheimer’s disease [57,58]. As these disorders have altered adult neurogenesis in the SGZ of the dentate gyrus, α7 nAChRs may control the normal maturation and integration of immature neurons and promote their survival [59,60,61,62]. Furthermore, Otto and Yakel found that blocking or removing α7 nAChRs increases neurogenesis overall but decreases NSC pools and special discrimination in adult males only, demonstrating the sexually dimorphic regulation of adult neurogenesis [63]. It is difficult to discern the impact of α7 nAChR activation on adult neurogenesis. The different and contradictory actions of this receptor may be due to the timing and location of its activation as well as a sexually dimorphic fashion [63,64]. Other α7 subunit-containing nAChRs, including the α7β2 subtype, are expressed in a diverse array of cells in the hippocampus, and their loss contributes to multiple neuropsychiatric and neurodegenerative disorders [65,66,67,68,69]. Thus, the regulation of adult neurogenesis via α7 subunit-containing nAChRs may provide a potential therapeutic strategy for treating neurodegenerative and neurological diseases. In the SGZ, immunohistochemical staining and functional analyses have revealed that type 1 and 4 mAChRs (M1 and M4) are expressed in immature hippocampal neurons [38,70]. Additionally, bromodeoxyuridine (BrdU) labeling analysis has shown that proliferating SGZ cells expressing M1 and M4 are also labeled with BrdU, suggesting the modification of NSC/progenitor cell populations [49].

mAChRs, which are metabotropic, seven-transmembrane proteins coupled to G proteins, activate various intracellular signaling pathways to control cellular function, including that of adult stem cells [23,70,71]. On the other hand, nAChRs are pentameric, ionotropic channels that mediate fast cholinergic transmission in the peripheral and central nervous systems [72]. Furthermore, the nAChR subtype, α2β4, is also involved in adult ISC function [73,74]. mAChR and nAChR signaling probably cooperates to fine-tune effects on cells including adult stem cells.

## 3. Epidermal Stem Cells in the Adult Skin

Mammalian skin provides an interface between organisms and their environments. The mammalian epidermis is composed of three self-renewing compartments: the hair follicle, sebaceous glands, and interfollicular epidermis [75]. Under normal conditions, the epidermises are maintained by their different stem cells [76,77]. When tissue homeostasis is disrupted by injury, each stem cell population is capable of producing each structure [78,79]. The importance of stem cell heterogeneity and compartmentalization via cholinergic signaling is discussed below.

### 3.1. Cholinergic Activation of Keratinization in the Skin

The skin is composed of two major layers, the epidermis and dermis. The epidermis is composed of a multilayered stratified epithelium that undergoes a specialized form of differentiation named keratinization [80]. The constantly renewing epithelial tissue is maintained by the proliferation of various epidermal stem cell populations that are located in the basal layer and their progressive differentiation after desquamation and injury [81]. Keratinocytes are not only tightly anchored to their neighboring cells via junctional complexes but also directly exchange information with one another in a manner that controls their differentiation state [81]. The cells have the ability to synthesize high amounts of ACh [82]. Both classes of ACh receptors are expressed in epidermal keratinocytes [83]. They also express members of the necessary transporters (choline transporter and VAChT) for an efficient autocrine and/or paracrine cholinergic loop, as well as the ACh degradation enzyme, AChE [82]. Thus, keratinocytes express the molecular components of the non-neuronal cholinergic system. The non-neuronal cholinergic system is postulated to regulate tight connections to other epidermal cells and/or stem cells, proliferation, differentiation, apoptosis, adhesion, and migration in the skin [81,83]. The regulated expression of cholinergic signaling in healthy skin may play a role in determining cell fate and positioning during epidermal wound healing, but this remains to be rigorously investigated.

### 3.2. Cholinergic Activation of Hair Follicle Stem Cells in the Skin

The hair follicle is a skin appendage that is contiguous with the epidermis. The cycle is divided into three stages: anagen (regeneration), catagen (degeneration), and telogen (rest) (Figure 4) [84]. This regeneration process relies on hair follicle stem cells (HFSCs) and melanocyte stem cells (MeSCs) in the bulge and hair germ region [85,86,87]. When HFSCs are activated, a new hair follicle is produced. *Lgr5* is a marker of HFSCs as well as a marker of stem cells in the intestine [88,89]. The gene is expressed in the bulge region and hair germ during long periods of time to contribute to all hair lineages [89]. A closely related gene, *Lgr6*, also contributes as a marker for all distinct populations of stem cells that give rise to all lineages of the skin [90]. The LGR family proteins can be divided into three groups: type A, B, and C [91]. LGR5 and 6 belong to type B [91]. In 2011 and 2012, R-spondins (RSpo1-4) were identified as endogenous ligands of these receptors [92,93,94,95,96,97]. Unlike *Lgr5*, *Lgr6* is not controlled by Wnt signaling [90]. The Wnt-independent Lgr6^+^ stem cell pool can renew sebaceous cells and seed the epidermis throughout life, whereas a Wnt-dependent Lgr5^+^ stem cell pool emerges from the Lgr6^+^ stem cell pool early in life and then becomes relatively independent from the Lgr6^+^ stem cell pool [90].

When MeSCs are activated, melanocytes differentiate, and mature cells migrate downwards. However, MeSCs remain close to the bulge. At the hair bulb, mature melanocytes synthesize melanin to color the newly regenerating hair from the root [98,99]. At the catagen stage, mature melanocytes vanish. Then, MeSCs initiate new rounds of melanogenesis in future cycles [98,99]. During aging, the pool of MeSCs is gradually depleted, and then pigmented hair becomes gray in color, and finally, white in color because of a complete loss of pigment in all hair follicles [99]. Psychological stress has been associated with changes from normal black hair to gray hair. However, little scientific validation of this link is available. Recently, Zhang and coworkers [100] revealed that signaling from the sympathetic nervous system in mice subjected to stress leads to depletion of the MeSC population in their hair follicles, and eventually hair graying. In addition to anti-graying therapeutics, their work will provide better understanding of how stress influences other stem cell pools and their niches such as ISCs and their niche.

Five mAChR subtypes (M1–M5) are expressed both in epidermal keratinocytes and melanocytes [101,102]. In particular, M4 is expressed at the highest level [101]. A role for M4 in keratinocyte migration was first revealed by comparing wound healing in M4 knockout (M4-KO) and WT mice. M4-KO mice show a significantly decreased epithelization rate that is concomitant with a reduced migration distance of keratinocytes [103,104]. In epidermal melanocytes, the regulation of intracellular free Ca^2+^ plays a fundamental role in the control of melanocyte dendricity [105]. The hypothesis is that M4 inhibits melanogenesis via the inhibition of adenylate cyclase and cyclic AMP synthesis because of selective coupling to G_i_ protein. This represents negative feedback regulation of the catecholamine/β2-adrenergic receptor response in melanocytes [106,107]. A recent study revealed that MeSCs express β2-adrenergic receptors, which respond to noradrenaline involved in the “fight or flight” response to stress [100]. Furthermore, deletion of this receptor specifically in MeSCs completely blocks stress-induced graying [100]. ACh is expected to be involved in MeSC homeostasis via M4 in the hair follicle cycle. A pilot study has been conducted for the M4-coupled pathway involved in murine hair follicle cycling and pigmentation [106]. When WT mice enter the first hair growth cycle, the M4-KO mice still show a slightly retarded catagen phase [106]. Subsequently, hair follicles of the M4-KO mice remain in a highly significantly prolonged telogen phase compared with that of WT mice [106]. Of interest, the M4-KO mice do not show follicular melanogenesis and fail to produce pigmented hair shafts [106]. Collectively, in a complex neuroectodermal–mesodermal interaction system, neuronal and/or non-neuronal ACh signaling leads to the depletion of stem cell populations in murine hair follicles via the M4 subtype. This discovery will increase understanding of human hair follicle biology.

## 4. ISCs in the Adult Gut

The ability of ISCs to divide and differentiate is necessary for tissue repair and homeostasis. The maintenance of a functional intestine requires appropriate spatial and temporal processes involving multiple key signals from the surrounding niche [108,109]. The ISC niche in the small intestine is composed of stem cells and Paneth cells, and it is surrounded by mesenchymal cells at the crypt bottom [110,111]. This provides a unique microenvironment that constitutes a constantly renewing dynamic system along the crypt–villus axis throughout postnatal life [112,113]. *Lgr5*-expressing ISCs temporarily produce undifferentiated cells that divide rapidly while moving toward the intestinal lumen [114]. During migration, these cells differentiate into mature cells such as goblet cells, tuft cells, enteroendocrine cells, and absorptive cells (enterocytes) [115,116,117]. Paneth cells move to the crypt bottom. Therefore, the densities of crypts and villi are dependent on ISC division [118]. In addition, B-cell-specific Moloney murine leukemia virus integration site 1 (Bmi1) labels quiescent ISCs, which are located at position +4 from the base of the crypt and have the ability to revert to Lgr5^+^ ISCs after damage to regenerate the villi [119,120]. Potten and coworkers [121] have provided experimental support for the +4 stem cell model for the first time. They have reported the existence of label-retaining cells residing specifically at this position [121]. These special mechanisms are important for life-long steady-state maintenance of the epithelium [111,122,123,124]. The influence of cholinergic signaling on ISC activity and differentiation is currently being elucidated.

### 4.1. Cholinergic Activation of ISCs via mAChRs

Muscarinic receptors (M1–M5) are G-protein-coupled receptors that mediate mucosal ion transport [125], epithelial proliferation [126], barrier function [127], and immune host defense mechanisms [128], as well as cholinergic neurotransmission at effector cells. M3, a receptor subtype that is expressed widely throughout the gastrointestinal tract, couples to G_αq/11_ to increase intracellular calcium via the activation of phospholipase C signaling and inositol phosphate formation [129]. Thus, M3 signaling alters cell function, including proliferation and differentiation [23,130]. Our previous pharmacological studies with crypt–villus organoids revealed that ACh is synthesized in intestinal epithelial cells and plays a role in cell division and the differentiation of Lgr5^+^ ISCs in the small intestine by binding to muscarinic receptors including M3 in vitro [23]. How ISC proliferation, differentiation, and maintenance are controlled and which inductive signals are required for tissue maintenance are well established [108]. However, little is known regarding the regulation of these pathways in vivo.

In the intestinal epithelium, tuft cells express ChAT for the production of ACh [131]. Intestinal tuft cells comprise a heterogeneous cell lineage and have been divided into two types, immune and neuronal phenotypes [117,132,133]. Tuft cells are chemosensory cells in the epithelial lining of the intestine. The brush-like microvilli projects from the cells. Similar tuft cells are found in the respiratory epithelium where they are known as brush cells [134]. Intestinal tuft cells have been proposed to be an important epithelial component of the ISC niche. After injury, the loss of tuft cells causes an impairment of intestinal regeneration [135]. Of interest, progenitors of tuft cells are located just above the crypt base at positions from +4 to +5, which is close to the position of Bmi1^+^ ISCs [136]. Enteroendocrine tuft cells may sustain quiescent Bmi1^+^ ISCs. Recently, Middelhoff and coworkers [109] revealed a novel interaction between M3 signaling and Lgr5^+^ ISC maintenance. They showed that the deletion of M3 reduces the number of Lgr5^+^ ISCs. Simultaneously, enteroendocrine tuft cells sense the M3 signaling interruption and upregulate epidermal growth factor signaling to sustain epithelial homeostasis [109]. ACh has been suggested to transactivate epidermal growth factor regulatory pathways through binding to M1, M2, and M3 [137,138,139,140]. Thus, the result is consistent with the signaling mechanisms underlying the proliferation and differentiation of Lgr5^+^ ISCs. The control of Lgr5^+^ ISCs is in part orchestrated by an increase in mucosal ACh release and is suggestive of a compensatory response circuit to maintain epithelial cholinergic input [109].

The identification of signaling pathways that have divergent effects in tissue stem/progenitor and cancer cells may offer insights into cancer development as well as offer novel therapeutic targets. M3 is interesting in this context, because genetic inhibition of M3 activity and treatment with the muscarinic receptor antagonist, scopolamine butylbromide, attenuate small intestinal adenoma formation in Apc^min/+^ mice [141]. Thus, the data so far obtained suggest an important role for M3 and ACh signaling in organizing ISC homeostasis in the gastrointestinal tract by governing proliferation in crypts and differentiation in villi. Loss of M3, which disrupts ISC homeostasis, appears to render the gut less vulnerable to tumorigenic progression.

### 4.2. Cholinergic Activation of ISCs via nAChRs

Nicotinic AChRs use ACh as an endogenous ligand and the agonist, nicotine, for signaling activation. nAChR signaling is a central regulator of physiological homeostasis in the central and peripheral nervous systems. These receptors also play a pivotal role in regulation of epithelial cell growth, migration, differentiation, and inflammation processes in various mammalian non-neuronal cells [73,82,142]. We have shown that nicotine increases organoid growth and differentiation [73]. In contrast, mecamylamine has an antagonistic effect compared with nicotine [73]. Our data obtained with immunohistochemical analysis represent the first description of the α2β4 subtype in Paneth cells in the crypt region and imply potentially novel functions such as the regulation of stem cell proliferation and differentiation [73]. RNA-Seq analysis has revealed that Wnt5a expression is dramatically upregulated after nicotine treatment, and recombinant Wnt5a rescues organoid growth and differentiation in response to mecamylamine [73]. Our results have indicated that coordinated activities of nAChR and Wnt signaling maintain Lgr5^+^ stem cell activity and balanced differentiation. Our RNA-Seq experiments also showed that differential expression levels of *Yes-associated protein* (*YAP1*), *transcriptional co-activator with PDZ-binding motif* (*TAZ*), *Notch receptor 1* (*Notch1*), and *delta-like ligand 1* (*Dll1*) are key aspects that are downstream of nAChR signaling [143]. These molecules are main components in Hippo and Notch signaling. The results suggest that the Hippo pathway and Notch signaling crosstalk with each other.

We also found that deficiency in the β4 subunit causes a decrease in crypt size and ISC proliferation and differentiation [143]. Our data suggest in part that the regulation of ISCs in normal adult mouse crypts is linked to the upregulation of nAChR-driven Hippo and Notch signaling pathways. Collectively, endogenous ACh binds to the α2β4 nAChR subtype in Paneth cells, which are critical components of the ISC niche both in vivo and in vitro [88,110]. Next, Hippo and Notch signaling pathways induce the expression of target genes such as *Wnts* (*Wnt5a* and/or *Wnt9b*) [73]. Wnts bind to various Frizzled receptors to activate Wnt signaling in ISCs, and eventually the proliferation and differentiation of ISCs are enhanced. Notch signaling, which involves the interaction between Notch1 and Dll1, in Paneth cells and ISCs modulates small intestinal homeostasis via the control of ISCs and induction of absorptive cells [110]. The non-neuronal cholinergic system probably, both functionally and independently, regulates or controls cell functions outside of the enteric nervous system in the mouse intestine.

Intestinal bowel disease (IBD), including Crohn’s disease and ulcerative colitis, is a chronic idiopathic inflammatory disorder that affects the gastrointestinal tract. Appropriate intestinal epithelial regeneration is required to improve the outcomes of treatment for IBD [74]. YAP1 regulation is a responsive mechanism for mucosal regeneration in an IBD mouse model induced using dextran sulfate sodium [108,144]. We suggest that the three-part mechanism of nAChR, Hippo, and Notch signaling permits adjustments in the rate of ISCs and their daughter cells to maintain the stem and daughter cell density in the adult mouse over the long term. Thus, upregulated nAChR signaling contributes to ISC function through the activation of Hippo signaling, which may be developed as a therapeutic target for IBD treatment.

In summary, the simultaneous stimulation of both ACh receptor classes may be required to synchronize and balance ionic and metabolic events in a single cell and/or tissue. Crosstalk between these receptors may fine-tune the signals emanating from epithelial cells and contribute to repair following tissue injury caused by inflammation. Signaling through the M3 muscarinic receptor and α2β4 nicotinic receptor appears to work together to maintain the homeostasis of intestinal epithelial cell growth and differentiation following modifications of the cholinergic intestinal niche (Figure 5). A systematic analysis of all components of the non-neuronal cholinergic system will increase our understanding of the cholinergic properties of non-neuronal cells and in turn lead to optimization of drug therapy.

## 5. Future Directions

Stem cells can divide extensively and have the ability to produce committed cells during early mammalian development. Early in development, stem cells need to be able to divide and differentiate into various lineages in response to external signals. Adult stem cells in tissues also have tremendous potential. They are maintained in a steady state in which each division typically produces one replacement stem cell and one tissue cell with no apparent limit [91]. The appropriate control of adult stem cells via cholinergic signaling may have potential anti-aging effects.

New neurons are generated from hippocampal NSCs that are located in the SGZ of the dentate gyrus. However, their generation is substantially diminished in aged animals due to a decrease in the NSC population. Voluntary exercise may increase neurogenesis in the hippocampus [145,146]. Additionally, Itou and coworkers [147] showed that cholinergic stimulation such as with donepezil can promote the proliferation of aged NSCs, which may lead to an increase in the number of new neurons in aged animals. The results support the conclusion that adult hippocampal neurogenesis may ameliorate the decline in cognitive functions that accompanies normal aging.

As people grow older, the pool of MeSCs is gradually depleted, pigmented hair becomes salt-and-pepper colored, and then turns to gray and finally to white after a complete loss of pigment in all hair follicles [99]. As M4 muscarinic receptor-mediated signaling is critical for normal hair follicle cycling and pigmentation in mice, clarifying why the defect in hair follicle pigmentation occurs and why a deficiency in the migration and/or differentiation of neural crest-derived precursor cells into the hair follicle occurs in M4-KO mice is important [106]. Answering these questions may in part lead to development of anti-graying therapies.

Nutrients are absorbed by the intestinal villi, and the absorption activity is affected by the size and density of villi [148]. Nutrient malabsorption is common among the elderly and often causes illness [149]. He and coworkers [150] showed that aging-induced intestinal villus structural and functional decline is regulated by mammalian target of rapamycin complex 1 (mTORC1), which is a sensor of nutrients and growth factors that is highly activated in ISC and progenitor cells in geriatric mice. Their findings revealed that mTORC1 drives aging by augmenting a prominent stress response pathway in ISCs and identified p38 mitogen-activated protein kinase as an anti-aging target downstream of mTORC1 [150]. Although the interaction between ACh receptors and mTORC1 remains unclear, therapy to alter ISC behavior for anti-aging as well as homeostasis and disease may involve the cholinergic intestinal niche.

## Figures and Tables

**Figure 1 ijms-22-00666-f001:**
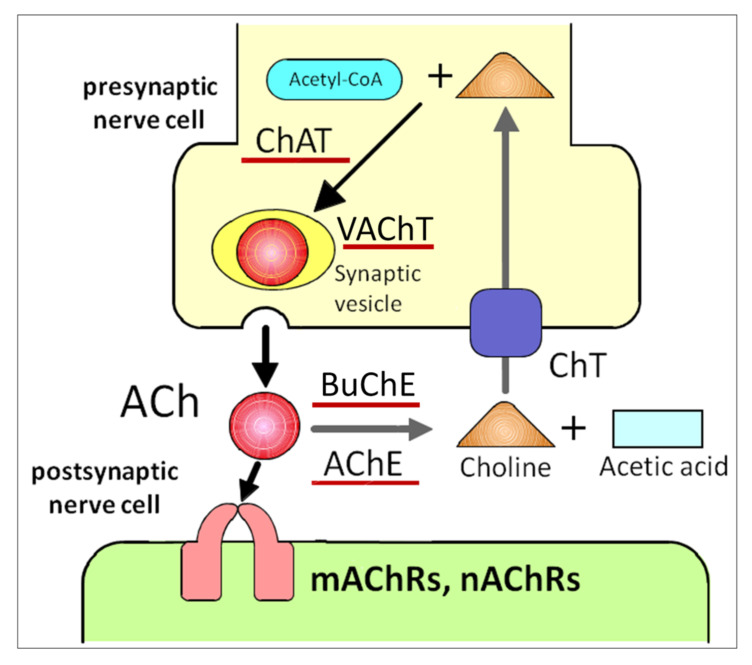
ACh release and receptor activation impacts neuronal activity. ACh is directly released into the synaptic cleft, followed by binding to nAChRs and mAChRs on the postsynaptic cell. Upon release, ACh is quickly degraded by extracellular AChE. ACh: acetylcholine, ChAT: choline acetyltransferase, VAChT: vesicular ACh transporter, mAChRs: muscarinic ACh receptors, nAChRs: nicotinic ACh receptors, AChE: acetylcholinesterase, BuChE: butyrylcholinesterase, ChT: choline transporter.

**Figure 2 ijms-22-00666-f002:**
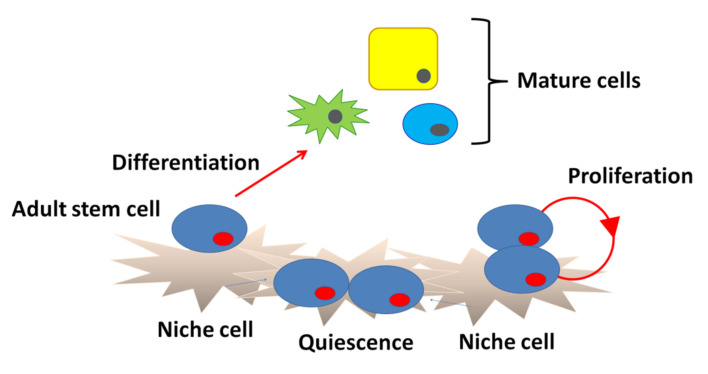
Niche structure. Niche cells under the basement membrane signal to stem cells to block differentiation and regulate division. Upon commitment, the stem cells differentiate into mature cells.

**Figure 3 ijms-22-00666-f003:**
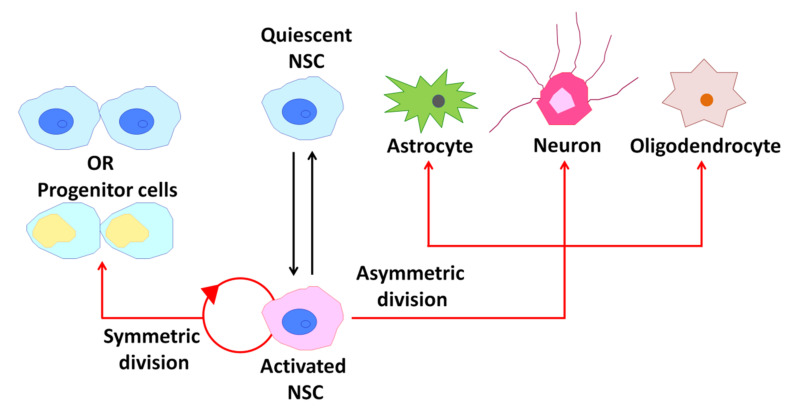
Behavior of neural stem cells (NSCs) within the adult mammalian brain. A schematic diagram illustrating the potential behavior of an NSC over its life cycle.

**Figure 4 ijms-22-00666-f004:**
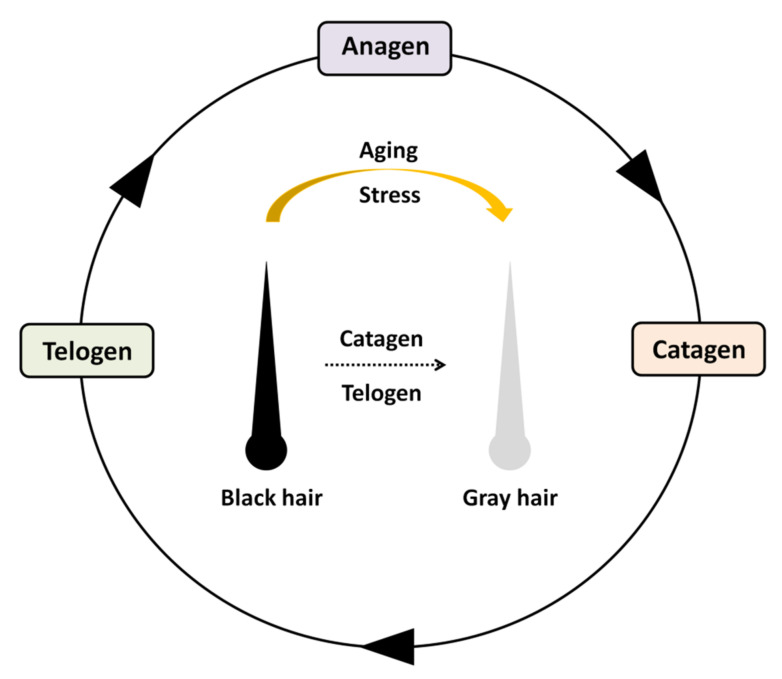
Hair cycle. The follicle cycles through three phases: anagen (regeneration), catagen (degeneration), and telogen (rest). The hair follicle is depleted of melanocyte stem cells (MeSCs) that would have differentiated to replace these melanocytes. Without any pigment cells to color the hair in the next anagen phase, the hair begins to look gray or white.

**Figure 5 ijms-22-00666-f005:**
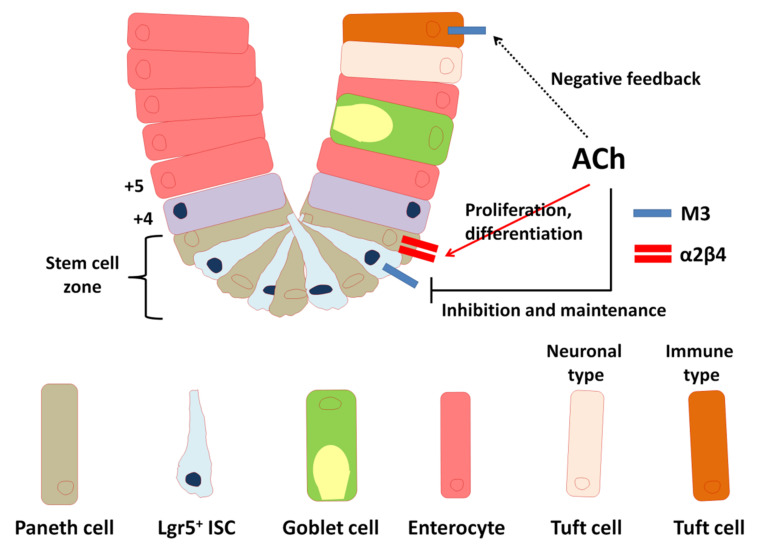
Model summarizing the importance of the cholinergic intestinal niche for maintenance of epithelial homeostasis. ACh: acetylcholine, M3: type 3 muscarinic ACh receptor, α2β4: nicotinic ACh receptor subtype composed of α2 and β4 subunits.

## Data Availability

Not applicable.

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
