# Peer review of "Multiple Roles for Cholinergic Signaling from the Perspective of Stem Cell Function"

_ijms, 2021, doi:10.3390/ijms22020666_

Round 1
Reviewer 1 Report
The review of Takahashi is summarizing the cholinergic control in stem cell function with a focus on the brain, skin, and gut. The manuscript is well written, and the topic is interesting. However, I have some remarks and suggestions.
Please add a appropriate reference in line: 127, 161, 163, 185, 241, 305.
Line 345 -352: In this section, the author is pointing out a possible interaction between ACh receptors and mTORC1. From my point of view, this new topic does not fit into the conclusion part of the manuscript as it is an unproven hypothesis and mTORC1 is not mentioned earlier in the manuscript. Moreover, while the term “aging” is only mentioned once in the manuscript (line 188), the whole conclusion section is about ”anti-aging”. Please rephrase or rearrange to improve the conclusion section.
Figure 5: Please define the abbreviations in the figure legend.
Author Response
Reviewer 1
The review of Takahashi is summarizing the cholinergic control in stem cell function with a focus on the brain, skin, and gut. The manuscript is well written, and the topic is interesting. However, I have some remarks and suggestions.
Comment: Please add a appropriate reference in line: 127, 161, 163, 185, 241, 305.
Answer: According to the reviewer's comment, I add a appropriate reference in the lines as follows. And, I changed the number of reference in the revised manuscript.
Line127: [63] Otto, S.L.; Yakei, J.L. The α7 nicotinic acetylcholine receptors regulate hippocampal adult-neurogenesis in a sexually dimorphic fashion. Brain Struct. Funct. 2019, 224, 829-846.
Line161: [83] Grando, S.A.; Pittelkow, M.R.; Schallreuter, K.U. Adrenergic and cholinergic control in the biology of epidermis: physiological and clinical significance. J. Invest. Dermatol. 2006, 126, 1948-1965.
Line163: [82] Grando, S.A.; Kist, D.A.; Qi, M.; Dahl, M.V. Human keratinicytes synthesize, secrete, and degrade acetylcholine. J. Invest. Dermatol. 1993, 101, 32-36.
Line185: [98] Nishimura, E.K.; Jordan, S.A.; Oshima, H.; Yoshida, H.; Osawa, M.; Moriyama, M.; Jackson, I.J.; Barrandon, Y.; Miyachi, Y.; Nishikawa, S. Dominant role of the niche in melanocyte stem-cell fate determination. Nature 2002, 416, 854-860.
Line 185: [99] Nishimura, E.K.; Granter, S.R.; Fisher, D.E. Mechanisms of hair graying: incomplete melanocyte stem cell maintenance in the niche. Science 2005, 307, 720-724.
Line 241: [129] Qin, K.; Dong, C.; Wu, G.; Lambert, N.A. Inactive-state preassembly of G(q)-coupled receptors and G(q) heterotrimers. Nat. Chem. Biol. 2011, 7, 740-747.
Line 305: [74] Takahashi, T. Roles of nAChR and Wnt signaling in intestinal stem cell function and inflammation. Int. Immunopharmacol. 2020, 81, 106260.
Comment: Line 345 -352: In this section, the author is pointing out a possible interaction between ACh receptors and mTORC1. From my point of view, this new topic does not fit into the conclusion part of the manuscript as it is an unproven hypothesis and mTORC1 is not mentioned earlier in the manuscript. Moreover, while the term “aging” is only mentioned once in the manuscript (line 188), the whole conclusion section is about ”anti-aging”. Please rephrase or rearrange to improve the conclusion section.
Answer: I agree with the reviewer's comment. In conclusion, I mainly described future directions concerning "anti-aging". Thus, this new topic dose not fit into the conclusion part of the manuscript. Accordingly, I changed the subtitle to "Future directions".
Comment: Figure 5: Please define the abbreviations in the figure legend.
Answer: According to the reviewer's comment, I added the abbreviations in Figure 5 as follows.
ACh: acetylcholine, M3: type 3 muscarinic ACh receptor, α2β4: nicotinic ACh receptor subtype composed of α2 and β4 subunits.
Reviewer 2 Report
This review paper handles the introduction of the non-neuronal cholinergic system in stem cell function, and provides us with comprehensive findings including the system being involved in self-renewal and differentiation.
However, in several parts of the review paper it needs some revisions in order to make it clearer and easier to be understood, as follows.
- Introduction line 41-43 page 1: The paper mentioned that ACh contributes to the Hydra nervous system; however, ACh has not been detected. For the referee, it seems strange. It should be described clearer.
2. Introduction line 54-57 page 2: The paper mentioned OCTs as a choline transporter. In contrast, CHT1 is also known as a choline transporter. Is there any differences between CHT1 and OCTs in terms of the specificity or selectivity of transporter? It needs more detailed explanations.
3. 2. NSCs in the adult mammalian brain line 123-127 page 3: The paper mentioned alpha7 nAChRs may promote neuronal survival; however, blocking this receptors increases neurogenesis but decreases NSC pool. Did this part mean that both stimulating and blocking the receptor may lead to the similar phenotype, i.e., enhancement of survival and neurogenesis? It should be explained more accurately why conflicting triggers can induce a common phenotype.
4. 3. cholinergic activation of hair follicle stem cells in the skin line 176-178 page 4: It needs more detailed explanation regarding Lgr5 and 6.
5. 4. ISCs in the adult gut line 230 page 5: "at position +4" is not familiar for readers, and therefore, it needs more detailed explanation.
6. line 250 page 6: It needs further explanation about "tuft cells".
Author Response
Reviewer 2
This review paper handles the introduction of the non-neuronal cholinergic system in stem cell function, and provides us with comprehensive findings including the system being involved in self-renewal and differentiation.
However, in several parts of the review paper it needs some revisions in order to make it clearer and easier to be understood, as follows.
- Comment: Introduction line 41-43 page 1: The paper mentioned that ACh contributes to the Hydra nervous system; however, ACh has not been detected. For the referee, it seems strange. It should be described clearer.
Answer: According to the reviewer's comment, I clearly described the sentences as follows.
Line41-43 page 1: It has been suggested that classical molecules such as acetylcholine (ACh) also contribute to the Hydra nervous system from the results of pharmacological experiments [6]. The membrane-bound ACh receptor and acetylcholinesterase have been demonstrated and confirmed by genome analysis [7,8]. Though ACh and other ACh receptor agonists function in neuronal and/or neuromuscular communication to regulate muscle contractions in Hydra, ACh itself has not been detected.
Comment: 2. Introduction line 54-57 page 2: The paper mentioned OCTs as a choline transporter. In contrast, CHT1 is also known as a choline transporter. Is there any differences between CHT1 and OCTs in terms of the specificity or selectivity of transporter? It needs more detailed explanations.
Answer: Thank you very much for the reviewer's comment. To describe the specificity and selectivity between CHT1 and OCTs in more detail, I added the following sentences.
Line54-57 Page2: (New sentences) ACh is stored at and released from VAChT in neurons [17]. Of interest, VAChT is expressed in cell-type specific manner in non-neuronal cells [18]. The cells that do not express VAChT have no ability to store ACh, but directly release ACh via OCTs [19,20]. Thus, OCTs are two in one choline and ACh transporters.
I added new references as follows.
[17] Erickson, J.D.; Varoqui, H.; Schafer, M.K.; Modi, W.; Diebler, M.F.; Weihe, E.; Rand, J.; Eiden, L.E.; Bonner, T.I.; Usdin, T.B. Functional identification of a vesicular acetylcholine transporter and its expression from a 'cholinergic' gene locus. J. Biol. Chem. 1994, 269, 21929-21932.
[18] Kummer, W.; Lips, K.S.; Pfeil, U. The epithelial cholinergic system of the airways. Histochem. Cell Biol. 2008, 130, 219-234.
[19] Lips, K.S.; Volk, C.; Schmitt, B.M.; Pfeil, U.; Arndt, P.; Miska, D.; Ermert, L.; Kummer, W.; Koepsell, H. Polyspecific cation transporters mediate luminal release of acetylcholine from bronchial epithelium. Am. J. Respir. Cell Mol. Biol. 2005, 33, 79-88.
[20] Koepsell, H.; Lips, K.; Volk, C. Polyspecific organic cation transporters: structure, function, physiological roles, and biopharmaceutical implications. Pharm. Res. 2007, 24, 1227-1251.
Comment: 3. 2. NSCs in the adult mammalian brain line 123-127 page 3: The paper mentioned alpha7 nAChRs may promote neuronal survival; however, blocking this receptors increases neurogenesis but decreases NSC pool. Did this part mean that both stimulating and blocking the receptor may lead to the similar phenotype, i.e., enhancement of survival and neurogenesis? It should be explained more accurately why conflicting triggers can induce a common phenotype.
Answer: I agree with the reviewer's comment. To explain more accurately why conflicting triggers can induce a common phenotype, I added new sentences as follows.
Line 123-127 Page3: (New sentences) It is difficult to discern the impact of α7 nAChR activation on adult neurogenesis. The different and contradictory actions of this receptor may be due to the timing and location of its activation, and also a sexually dimorphic fashion [63,64].
[64] Gu, Z.; Yakel, J.L. Timing-dependent septal cholinergic induction of dynamic hippocampal synaptic plasticity. Neuron 2011, 71, 155-165.
Comment: 4. 3. cholinergic activation of hair follicle stem cells in the skin line 176-178 page 4: It needs more detailed explanation regarding Lgr5 and 6.
Answer: According to the reviewer's comment, I added new sentences and references to explain Lgr5 and 6 as follows.
Line176-178 Page4: (New sentences) The LGR family proteins can be divided into three groups: type A, B, and C [130]. LGR5 and 6 belong to type B [130]. In 2011 and 2012, R-spondins (RSpo1-4) were identified as endogenous ligands of these receptors [92-97].
New references
[92] Carmon, K.S.; Gong, X.; Lin, Q.; Thomas, A.; Liu, Q. R-spondins function as ligands of the orphan reeptors LGR4 and LGR5 to regulate Wnt/β-catenin signaling. Proc. Natl. Acad. Sci. USA 2011, 108, 11452--11457.
[93] Carmon, K.S.; Lin, Q.; Gong, X.; Thomas, A.; Liu, Q. LGR5 interacts and co-internalizes with Wnt receptors to modulate Wnt/β-catenin signaling. Mol. Cell Biol. 2012, 32, 2054-2064.
[94] de Lau, W.; Barker, N.; Low, T.Y.; Koo, B-K.; Li, V.S.W.; Teunissen, H.; Kujala, P.; Haegebarth, A.; Peters, P.J.; van de Wetering, M.; et al. Lgr5 homologues associate with Wnt receptors and mediate R-spondin signalling. Nature 2011, 476, 293-297.
[95] Glinka, A.; Dolde, C.; Kirsch, N.; Huang, Y-L.; Kazanskaya, O.; Ingelfinger, D.; Boutros, M.; Cruciat, C-M.; Niehrs, C. LGR4 and LGR5 are R-spondn receptors mediating Wnt/β-catenin and Wnt/PCP signalling. EMBO Rep. 2011, 12, 1055-1061.
[96] Ruffner, H.; Sprunger, J.; Charlat, O.; Leighton-Davies, J.; Grosshans, B.; Salathe, A.; Zietzling, S.; Beck, V.; Therier, M.; Isken, A.; et al. R-spondin potentiates Wnt/β-catenin signaling through orphan receptors LGR4 and LGR5. PLoS One 2012, 7, e40976.
[97] Gong, X.; Carmon, K.S.; Lin, Q.; Thomas, A.; Yi, J.; Liu, Q. LGR6 is a high affinity receptor of R-spondins and potentially functions as a tumor suppressor. PLoS One 2012, 7, e37137.
Comment: 5. 4. ISCs in the adult gut line 230 page 5: "at position +4" is not familiar for readers, and therefore, it needs more detailed explanation.
Answer: According to the reviewer's comment, I explained "at position +4" in more detail as follows.
Line230 Page5: (New sentences) Potten and coworkers [121] have provided experimental support for the +4 stem cell model for the first time. They have reported the existence of label-retaining cells residing specifically at this position [121].
I added a new reference as follows.
[121] Potten, C.S.; Kovacs, L.; Hamilton, E. Continuous labelling studies on mouse skin and intestine. Cell Tissue Kinet. 1974, 7, 271-283.
Comment: 6. line 250 page 6: It needs further explanation about "tuft cells".
Answer: According to the reviewer's comment, I added new sentences and a reference to explain tuft cells as follows.
Line 250 Page6: (New sentences) Tuft cells are chemosensory cells in the epithelial lining of the intestine. The brush-like microvilli projects from the cells. Similar tuft cells are found in the respiratory epithelium where they are known as brush cells [134].
[134] Gerbe, F.; Jay, P. Intestinal tuft cells: epithelial sentinels linking luminal cues to the immune system. Mucosal Immunol. 2016, 9, 1353-1259.